# C-Terminal Extensions of Ku70 and Ku80 Differentially Influence DNA End Binding Properties

**DOI:** 10.3390/ijms21186725

**Published:** 2020-09-14

**Authors:** Takabumi Inagawa, Thomas Wennink, Joyce H. G. Lebbink, Guido Keijzers, Bogdan I. Florea, Nicole S. Verkaik, Dik C. van Gent

**Affiliations:** 1Department of Molecular Genetics, Erasmus MC, University Medical Centre, 3015 GD Rotterdam, The Netherlands; daisukemana@hotmail.com (T.I.); thomas_wennink@hotmail.com (T.W.); j.lebbink@erasmusmc.nl (J.H.G.L.); guido@sund.ku.dk (G.K.); b.florea@chem.leidenuniv.nl (B.I.F.); n.verkaik@erasmusmc.nl (N.S.V.); 2Department of Radiation Oncology, Erasmus MC, University Medical Centre, 3015 GD Rotterdam, The Netherlands

**Keywords:** DNA repair, non-homologous end-joining, DNA-PK, DNA binding, surface plasmon resonance (SPR)

## Abstract

The Ku70/80 heterodimer binds to DNA ends and attracts other proteins involved in the non-homologous end-joining (NHEJ) pathway of DNA double-strand break repair. We developed a novel assay to measure DNA binding and release kinetics using differences in Förster resonance energy transfer (FRET) of the ECFP-Ku70/EYFP-Ku80 heterodimer in soluble and DNA end bound states. We confirmed that the relative binding efficiencies of various DNA substrates (blunt, 3 nucleotide 5′ extension, and DNA hairpin) measured in the FRET assay reflected affinities obtained from direct measurements using surface plasmon resonance. The FRET assay was subsequently used to investigate Ku70/80 behavior in the context of a DNA-dependent kinase (DNA-PK) holocomplex. As expected, this complex was much more stable than Ku70/80 alone, and its stability was influenced by DNA-PK phosphorylation status. Interestingly, the Ku80 C-terminal extension contributed to DNA-PK complex stability but was not absolutely required for its formation. The Ku70 C-terminal SAP domain, on the other hand, was required for the stable association of Ku70/80 to DNA ends, but this effect was abrogated in DNA-PK holocomplexes. We conclude that FRET measurements can be used to determine Ku70/80 binding kinetics. The ability to do this in complex mixtures makes this assay particularly useful to study larger NHEJ protein complexes on DNA ends.

## 1. Introduction

DNA double-strand breaks (DSBs) are among the most genotoxic DNA lesions. They can be induced by ionizing radiation or certain chemicals, but they are also part of normal cellular processes, such as V(D)J recombination in lymphocytes. Correct repair of these breaks is very important to prevent chromosomal deletions and translocations, which can cause cancer or cell death [1]. The two major DSB repair pathways that deal with these lesions are non-homologous end-joining (NHEJ) and homologous recombination (HR). HR is mainly involved in the repair of replication-associated DSBs in the S phase of the cell cycle, whereas NHEJ probably has a role in all cell cycle phases [2,3].

NHEJ requires the assembly of protein-DNA complexes that contain several proteins [4]. The Ku70/80 heterodimer binds to DNA ends with high affinity and specificity. The protein forms a ring that can thread onto a DNA end, which explains its inability to bind to circular DNA molecules or internal positions on the chromosome [5]. Once bound to DNA, Ku70/80 attracts the catalytic subunit of the DNA-dependent protein kinase (DNA-PK_CS_). This interaction is mediated by the Ku80 C-terminus and other contact points between the Ku heterodimer and DNA-PK_CS_ [6,7,8,9,10,11,12]. Upon juxtaposition of two DNA ends, DNA-PK mediates autophosphorylation, which enables DNA end processing by polynucleotide kinase, Artemis nuclease, DNA polymerase, or other modifying factors [4,13,14,15]. The final step in the NHEJ reaction is ligation of the juxtaposed DNA ends by the ligase IV/XRCC4 complex, which also requires the XLF/Cernunnos protein and/or PAXX proteins [16].

Binding of Ku70/80 heterodimers to DSBs in mammalian cells occurs within a few seconds after DNA damage induction [17,18]. The dynamics of this process have been studied in living cells using green fluorescent protein (GFP)-tagged Ku80 protein. A detailed investigation into the dynamics of this process has revealed that the binding of Ku70/80 to DSBs is a reversible process, suggesting that NHEJ complexes can fall apart without accomplishing the final joining step. A better understanding of the factors that influence the stability of protein-DNA complexes involved in NHEJ is of importance to predict the outcome of a DSB repair reaction: disassembly of an NHEJ complex before completion of repair may increase the chances for chromosomal deletions and translocations. The inability to reverse the initial binding reaction, on the other hand, could interfere with alternative repair processes that may need to gain access to the ends (e.g., homologous recombination in recently replicated DNA). The importance of maintaining the balance between different DSB repair pathways is exemplified by several chromosomal instability syndromes, such as Ataxia–Telangiectasia and hereditary breast cancer [19]. Therefore, we studied the dynamics of NHEJ complex assembly in more detail using surface plasmon resonance (SPR) and Förster resonance energy transfer (FRET) in fluorescent versions of the Ku70/80 heterodimer in vitro. We further defined the function of the Ku70 and Ku80 C-terminal domains in DNA end binding and DNA-PK_CS_ interaction.

## 2. Results

### 2.1. The Enhanced Cyan Fluorescent Protein (ECFP)-Ku70/EYFP-Ku80 FRET Pair

The Ku70/80 heterodimer binds to DNA ends as a starting point for NHEJ complex assembly and juxtaposition of DNA ends. As this is a highly dynamic process [17], we developed a time-resolved DNA binding assay based on fluorescence. We used FRET to probe the distance and relative orientation between the fluorophores—enhanced cyan fluorescent protein (ECFP) and enhanced yellow fluorescent protein (EYFP). We tagged Ku70 with ECFP and Ku80 with EYFP (both on the N-terminus), produced the ECFP-Ku70/EYFP-Ku80 heterodimer in a Baculovirus expression system, and purified it to near homogeneity (Figure 1D and Appendix A). The double-tagged heterodimer was able to bind to DNA ends (Appendix A) and form a complex with DNA ligase IV/XRCC4 (Appendix A). The tags did not interfere with DNA-PK_CS_ activation, and ECFP-Ku70 and EYFP-Ku80 were able to complement the genetic defect in Ku70 and Ku80 deficient cells, respectively ([9,17] and Appendix A), showing that the tagged Ku70 and Ku80 proteins could be used for functional studies on Ku protein behavior. 

The purified ECFP-Ku70/EYFP-Ku80 protein showed an efficient FRET signal (Appendix A), which was abrogated upon disruption of the heterodimer by lithium dodecyl sulfate (LDS). The signal was sensitive to salt concentration, temperature, and pH (Appendix A). The FRET efficiency of the heterodimer was optimal around pH 8.0, salt concentrations of 120–150 mM KCl, and a temperature of 25 °C. Therefore, we chose to do the subsequent incubations at 25 °C, pH 8.0, and 120 mM KCl, which was also close to physiological conditions. The FRET efficiency was calculated by subtracting the ECFP signal from the spectrum. For all reactions, we calculated the FRET efficiency relative to FRET efficiency under standard conditions, which was normalized to 100%. A mixture of ECFP-Ku70/Ku80 and Ku70/EYFP-Ku80 did not give a FRET signal, indicating that close proximity of the fluorophores in the same heterodimer complex was required for energy transfer [20].

### 2.2. DNA Ends Decrease the FRET Efficiency of ECFP-Ku70/EYFP-Ku80

We measured the effect of DNA binding on the ECFP-Ku70/EYFP-Ku80 FRET efficiency in order to evaluate its possible use for the real-time measurement of Ku-DNA binding. To prevent possible complications caused by the binding of multiple Ku molecules from both ends of the DNA fragment, we used biotinylated oligonucleotides with streptavidin to block access from one end (Figure 1A). Surprisingly, the addition of linear DNA fragments resulted in a dramatic reduction of the FRET efficiency (Figure 1B,C). The ECFP-Ku70/EYFP-Ku80 heterodimer interacted with blunt-ended (bl1), 5′ single-stranded extensions (ov1), and a DNA hairpin (hp1). We observed a similar decrease in the FRET signal with these three different dsDNA substrates. In contrast, the FRET signal decreased only slightly when we used oligonucleotide substrates with a terminus that could not be bound by Ku70/80 (substrate cv1, which we termed a ‘clover-leaf’ structure) [5]. The presence of a single-stranded break in the DNA (substrates hp2 and cv2) did not significantly alter binding characteristics, showing that only double-strand DNA breaks caused the change in the FRET signal.

### 2.3. Dissociation of ECFP-Ku70/EYFP-Ku80 from DNA Ends

In order to monitor the release of ECFP-Ku70/EYFP-Ku80 from DNA in the FRET assay, we added excess purified untagged Ku70/80, which could replace the tagged protein when it dissociates from the DNA terminus. Figure 2A shows that the FRET signal can be regained over an incubation period of three hours after the addition of untagged Ku70/80 to reaction mixtures containing the 5′ overhang substrate ov1 (dissociation rate constant *k*_off_ ov1 = 2.5 × 10^−4^ s^−1^). Interestingly, exchange was three times faster from the hairpin substrate hp1 (*k*_off_ hp1 = 8.5 × 10^−4^ s^−1^), while recovery for the blunt-ended substrate bl1 was only partial after three hours (*k*_off_ bl1 = 1.4 × 10^−4^ s^−1^), suggesting that Ku70/80 bound with quite different characteristics to these three types of DNA ends under near-physiological conditions. Interestingly, this striking difference was not observed when the salt concentration was increased to 250 mM KCl (Figure 3B): all substrates with accessible DNA ends showed recovery of the FRET signal within 15 minutes in the higher salt buffer, with only slightly faster dissociation for the hairpin substrate (*k*_off_ bl1 = 2.0 × 10^−3^ s^−1^; *k*_off_ ov1 = 2.0 × 10^−3^ s^−1^; *k*_off_ hp1 = 3.0 × 10^−3^ s^−1^).

### 2.4. Validation of Ku-DNA Binding Using SPR Analysis

To correlate the observed change in FRET efficiency to DNA binding, we analyzed the interaction of Ku70/80 and ECFP-Ku70/EYFP-Ku80 with the same 30-bp DNA substrates using surface plasmon resonance. At physiological ionic strength (120 mM), Ku70/80 (Figure 3A) and ECFP-Ku70/EYFP-Ku80 (Figure 3B) bound in a similar manner to the blunt and hairpin DNA substrates immobilized on the chip surface, while the cloverleaf did not show much binding of either protein. The only obvious difference between tagged and wild type Ku70/80 was a higher response for the labeled protein because of its higher mass due to the fluorescent fusion partners. Despite dissociation being incomplete, we estimated dissociation rate constants of 2.4 × 10^−4^ and 2.8 × 10^−4^ s^−1^ for Ku70/80 and ECFP-Ku70/EYFP-Ku80 from the blunt DNA (bl1). For the hairpin substrate hp1, these values were 4.6 × 10^−4^ and 6.7 × 10^−4^ s^−1^, respectively. This further supported our conclusion that the fluorescent fusion proteins did not influence the Ku70/80 function. Furthermore, the dissociation rate constants derived from SPR were similar to those observed in the FRET experiments on these substrates (for example, 2.8 × 10^−4^ s^−1^ for substrate bl1 compared to 1.4 × 10^−4^ s^−1^ for the FRET assay).

This analysis was repeated at 250 mM salt concentration to allow full dissociation. Again, we obtained very similar affinities for Ku70/Ku80 and ECFP-Ku70/EYFP-Ku80 (Figure 3C,D; Appendix A; Table 1). Furthermore, the association and dissociation rate constants were basically identical to previously reported values [22]. Ku70/80 bound with similar affinities and rate constants to substrates bl1 and ov1 (Table 1). However, the affinity for the hp1 substrate was significantly lower. This was caused by a large increase in the dissociation rate constant, which more than compensated the slightly increased association rate constant. Taken together, we concluded that the observed changes in FRET reported DNA binding and release by the fluorescently-tagged Ku protein.

### 2.5. Kinetics of Binding and Dissociation from DNA for Wild-Type and Mutant Ku

Although DNA binding is mainly determined by the protein ring encircling the DNA double helix, the Ku70 C-terminal extension (the SAP domain) has been shown to be able to bind DNA and probably contribute to its DNA binding properties, as well [21,23,24]. Therefore, we purified the ECFP-Ku70ΔSAP/EYFP-Ku80 protein and investigated its binding properties. In the FRET assay, we observed that the association was similar to wild-type Ku heterodimers, but its release was much faster (Figure 2C; 120 mM KCl). 

This was compared to SPR results, which were conducted at 250 mM KCl to allow full monitoring of the dissociation process. The ΔSAP variant was highly impaired in DNA binding measured by both SPR and FRET (Appendix A and Table 1). The ΔSAP variant had a slightly reduced association rate constant for the blunt substrate but very rapidly dissociated, resulting in an almost 20-fold reduced affinity for this substrate.

The Ku80 C-terminus had been found to influence mainly interaction with DNA-PKcs. Indeed, for the Ku80 C-terminal deletion mutant, we did not observe such a large difference in DNA binding characteristics. In the FRET assay at 120 mM KCl, the association was somewhat slower than for wild-type (Figure 2D), but this difference was not visible at 250 mM KCl. Dissociation was not severely affected by this mutation, although some differences were apparent for the ov1 substrate. SPR analysis revealed that the Ku70/80ΔC variant had a slightly reduced affinity for the blunt DNA substrate due to an approximately two-fold increase in the dissociation rate constant (Appendix A; Table 1).

### 2.6. DNA-PK Complexes with Mutant Ku Proteins

Subsequently, we investigated the effects of DNA-PK_CS_ on Ku70/80 binding to DNA ends. As changes in FRET efficiency could be used to monitor Ku-DNA end binding in complex mixtures, we investigated the effect of DNA-PK_CS_ on exchange kinetics from DNA ends using the FRET assay. As shown in Figure 4A, Ku exchange was virtually abrogated in the presence of DNA-PK_CS_, even at 250 mM KCl. As the stability of these complexes had been found to decrease after DNA-PK_CS_ autophosphorylation, we then added ATP to allow activation of its kinase activity. Indeed, Ku protein dissociation was increased considerably. The addition of the DNA-PK inhibitor Wortmannin to this reaction prevented Ku exchange, showing that this phenomenon was indeed dependent on DNA-PK activity. 

Subsequently, we investigated the effects of Ku mutations on DNA-PK complex stability. The Ku80 C-terminus was previously found to interact with DNA-PK_CS_, suggesting that DNA-PK complexes might not form with this mutant protein [6,8]. However, in vivo measurements showed that DNA-PK_CS_ could still accumulate in cells expressing a YFP-tagged form of this protein [9]. Consistent with both observations, we reported here that ECFP-Ku70/EYFP-Ku80ΔC, in combination with the DNA-PK_CS_ subunit, formed a DNA-PK complex that was more stable than the Ku heterodimer alone, although not as stable as the DNA-PK complex with wild-type Ku heterodimer (Figure 4B).

Surprisingly, we found that the Ku70 SAP domain did not influence the dissociation kinetics in the context of a complex with DNA-PK_CS_ (Figure 4B) in contrast to its faster dissociation in the absence of DNA-PK_CS_ (Figure 2C and Table 1). The ability of this mutant protein to bind in the context of a full DNA-PK complex suggested that it might function relatively normally in NHEJ in vivo. Therefore, we investigated its activity using a V(D)J recombination assay that is fully dependent on Ku function. As the Ku70ΔSAP mutant lacked a nuclear localization signal (NLS), we added a new NLS at the N-terminus to ensure proper subcellular localization. We indeed found that this mutant complemented the defect in Ku70-deficient mouse embryonic fibroblasts (Figure 4C). The Ku80 C-terminal deletion mutant, however, could not substitute for wild-type Ku80, as observed previously (Figure 4C).

## 3. Discussion

We developed a novel analysis tool to measure the dynamics of Ku70/80 binding to DNA ends, making use of DNA-dependent changes in FRET efficiency between ECFP and EYFP that were coupled to the N-termini of Ku70 and Ku80, respectively. The DNA-dependent changes of FRET efficiency correlated with the binding to and release from DNA with different structures analyzed by SPR. The Ku70 SAP domain was required for the efficient retention of Ku70/80 on DNA ends, whereas the Ku80 C-terminal domain stabilized the DNA-PK holocomplex on DNA ends.

Measurements using SPR and FRET have complementary advantages for different situations. The sensitivity of SPR is superior to FRET measurements when measuring the binding of a single protein species to a DNA substrate and does not require the presence of competitor protein during the dissociation phase, as it is monitored in a microfluidic flow system. However, measurement of FRET efficiency is advantageous when complex mixtures consisting of multiple proteins are investigated: SPR measures the total mass of protein binding to the DNA end, while FRET measurements visualize specifically binding of Ku70/80 in the complex mixtures, and this technology may even bridge the gap between in vitro binding reactions and in vivo measurements. For complex reactions, such as assembly of NHEJ complexes consisting of a combination of several proteins, SPR measurements are less suitable since it will be very difficult, if not impossible, to deconvolute the sensorgrams into the mass increase for the individual components.

Our analysis provides interesting insights into DNA end specificity. The clearest difference between the binding of Ku to different ends was found at physiological salt concentrations (120 mM KCl). FRET-monitored release from blunt DNA ends was clearly different in comparison with 5′ protrusions at 120 mM KCl, while this difference was not observed at 250 mM, suggesting that fundamentally different binding modes were operational under both conditions. It is not clear at present what causes this difference, but it highlights that observations under high salt conditions cannot be translated directly to the more physiological situation. 

Interestingly, DNA hairpin structures were bound with lower affinity than open DNA ends. More specifically, the dissociation rate constant was considerably higher for hairpin ends. The binding observed in both SPR and FRET experiments is definitely not aspecific binding to internal DNA sequences since the clover-leaf structure shows very different binding characteristics. DNA hairpins are the natural product of DNA cleavage at recombination signal sequences by the RAG proteins during V(D)J recombination in the immune system [25]. The RAG proteins normally cleave the DNA only when two recombination signal sequences are synapsed. However, DNA is sometimes also cleaved without the juxtaposition of both ends. In such a case, the integrity of the DNA can be restored by reversing the cleavage reaction, called open-and-shut joint formation. This reaction can be catalyzed by the RAG proteins without the need for NHEJ factors [26]. Obviously, such a reaction would only work when the DNA end is not occupied by the Ku heterodimer. Therefore, this reduced binding affinity may be important to prevent irreversible binding of Ku to the coding-end during V(D)J recombination, which might otherwise be harmful if DNA breaks occur at non-synapsed recombination signal sequences. This would be even more harmful if RAG proteins cleave at so-called cryptic recombination signal sequences that might give rise to translocations if coupled to an unrelated another DNA end.

DNA-PK_CS_ binding to a Ku-DNA complex rendered these complexes much more stable. The Ku heterodimer hardly dissociated anymore from blunt DNA ends, unless autophosphorylation led to a conformational change and less tight DNA end binding. DNA-PK complexes on blunt DNA ends with the Ku80 C-terminal deletion mutant were less stable than wild-type complexes but more stable than Ku without DNA-PKcs. This observation could probably explain why the Ku80 C-terminus had been reported to be necessary for DNA-PK_CS_ binding to DNA breaks in chromatin measured by a biochemical assay, while attraction to laser-induced DNA damage was relatively normal. Our observations clearly showed that DNA-PK complex formation was still possible, but the complexes had a lower affinity (Figure 4). In combination with reduced DNA-PK autophosphorylation activity, this may explain the inability of this mutant Ku80 to complement the DNA repair defect in Ku80 deficient cells [6,8,9].

## 4. Materials and Methods

### 4.1. Oligonucleotide Substrates

Biotinylated oligonucleotides were synthesized and PAGE purified by Eurogentec (Seraing, Belgium). Double-stranded oligonucleotide substrates were made by annealing of complementary strands (Figure 1A). Covalently closed hairpin and clover-leaf substrates were made by ligation of hp2 and cv2, respectively. The ligation products were purified from 12% polyacrylamide gels in TBE buffer with 8 M urea and 10% formamide. Purified samples were adjusted to 1 μM in TE buffer. For FRET assays, 1 μg streptavidin was added per pmol of oligonucleotide substrate, and mixtures were incubated for 10 minutes at room temperature.

### 4.2. Protein Purification

The expression and purification of Ku and Ku mutant heterodimers in a Baculovirus expression system were performed, as described [9]. ECFP-Ku70ΔSAP was generated by the deletion of codons 536 to 609 of the Ku70 open reading frame in the ECFP-Ku70 expression vector. Ligase IV/XRCC4 protein was expressed in *Escherichia coli* and purified, as described [13]. DNA-PK_CS_ was purified from HeLa cells, as described [13]. 

### 4.3. Electrophoretic Mobility Shift Assay (EMSA)

EMSA was performed, as described [9]; 0.2 pmol dsDNA (50 base pairs; DAR39/40) [25] was incubated for 15 minutes at room temperature with increasing Ku protein concentrations. Reaction mixtures were separated on 6% polyacrylamide gels in 0.5 × TBE at 50 V, 4 °C. Where indicated, ligase IV/XRCC4 (100 ng) was added to the binding reactions.

### 4.4. V(D)J Recombination Assay

Ku70^−/−^/Ku80^−/−^ mouse embryonic fibroblasts (kind gift of Dr. D. Chen) [27,28] were transfected with the V(D)J recombination substrate pDVG93 [29], RAG1 and RAG2 expression plasmids, and the EYFP-Ku80 expression plasmid pHB25 and/or the ECFP-Ku70 expression plasmid pGK49. Instead of wild-type expression constructs, the mutant expression constructs for pEYFPKu80ΔC or pECFP-Ku70ΔSAP (in which an additional nuclear localization signal was added at the N-terminus of the ECFP open reading frame) were used when indicated. After two days, extrachromosomal DNA was extracted, and signal joints were amplified using primers DAR5 (5′-TGCTTCCGGCTCGTATGTTGTGTGGAAT) and NV07R (5′-TCGCAAATTGTCGCGGCGATTAAATCTC), as described [29].

### 4.5. FRET

Fluorescence was measured with an LS50B Luminescence Spectrometer (Perkin Elmer, Waltham, MA, USA) or a FluoroMax-4 Spectrofluorimeter (Horiba Scientific, Kyoto, Japan) at 25 °C. The mixture was excited at 435 nm, and emission was recorded between 440 and 650 nm. One microgram of ECFP-Ku70/EYFP-Ku80 (or a Ku mutant protein) was incubated for 60 minutes in 80 µL FRET buffer (25 mM Tris-HCl pH8.0, 120 mM KCl, 10 mM MgCl_2_, and 2 mM DTT). DNA oligonucleotide substrates were prepared in FRET buffer containing 2 μg streptavidin per pmol gel-purified biotinylated oligonucleotides. Two pmol of the oligonucleotide with streptavidin was added to the Ku mixtures, and the fluorescent signals were measured every 15 minutes for 2 hours. Subsequently, 3 µg of non-tagged Ku70/80 was added, and the fluorescence was monitored every 15 minutes for 3 hours. Similar measurements were carried out in buffer containing 250 mM KCl instead of 120 mM. In the latter case, the fluorescence was measured every 5 minutes for one hour after the addition of non-tagged Ku70/80.

The relative FRET efficiency was calculated using a series of normalization steps. First, all spectra were normalized on the ECFP peak at 475 nm. Subsequently, the ECFP spectrum of ECFP-Ku70/Ku80 was subtracted from this FRET spectrum. The height of the resulting peak at 487 nm (the EYFP emission maximum) was divided by the 475 nm signal, and the relative FRET efficiency of ECFP-Ku70/EYFP-Ku80 was normalized to 100%. All values were calculated relative to this value of double-tagged Ku heterodimer without DNA. Dissociation rate constants were estimated by fitting a function describing a single exponential increase to the dissociation phase of the FRET experiments using Graphpad Prism version 8.3.1 (GraphPad Software, San Diego, CA, USA).

### 4.6. Surface Plasmon Resonance

SPR spectroscopy was performed at 25 °C on a Biacore T100 (GE Healthcare, Boston, MA, USA), as described [30]. Streptavidin SA sensor chips were derivatized with 10 resonance units of biotinylated DNA oligonucleotides. Ku70/80 (0.625–10 nM), ECFP-Ku70/EYFP-Ku80 (1.25–20 nM), or mutant variants ECFP-Ku70/EYFP-Ku80ΔC (3.1–50 nM), ECFP-Ku70ΔSAP/EYFP-Ku80 (6.3–100 nM), ECFP-Ku70ΔSAP/EYFP-Ku80ΔC (6.3–100 nM) in 25 mM Tris-HCl pH 8.0, 250 or 120 mM NaCl, 10 mM MgCl_2_, 2 mM DTT, 0.01% Surfactant P20 (GE Healthcare, Boston, MA, USA) were injected across the chip at 50 µL/min. The chip surface was regenerated with 0.1 % SDS. Dissociation rate constants for data at 120 mM salt were estimated by fitting a function describing a single exponential decay to the dissociation phase of the sensorgram using Graphpad Prism 8. Data at 250 mM salt were analyzed using a two-state binding model using BiaEvaluation software (GE Healthcare, Boston, MA, USA) for accounting for a fraction of bound Ku that interacted with the chip matrix, specifically on DNA-modified surfaces (Appendix A), and visualized using Graphpad Prism version 8.3.1 (GraphPad Software, San Diego, CA, USA).

## Figures and Tables

**Figure 1 ijms-21-06725-f001:**
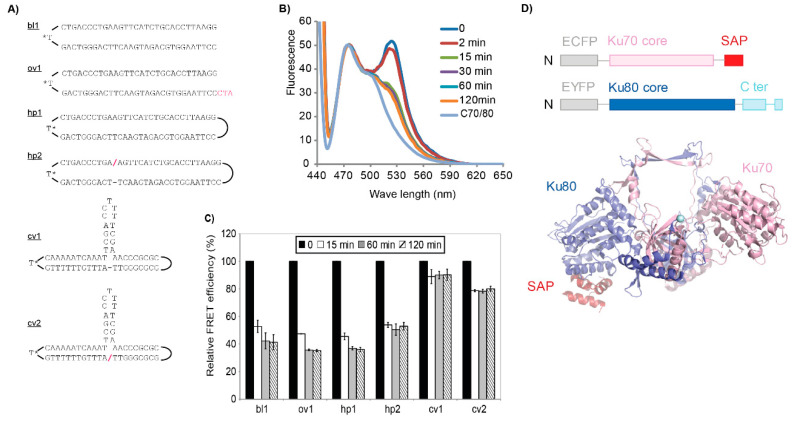
FRET (Forster resonance energy transfer) efficiency decreases upon Ku-DNA binding. (**A**) Schematic of the substrates; the difference between substrates hp1/hp2 and cv1/cv2 is the presence of a nick in substrates hp2 and cv2 (depicted by a red/-sign); T* = biotinylated T. (**B**) FRET efficiency (525 nm peak) decreased in time upon addition of substrate ov1. C70/80 = ECFP (enhanced cyan fluorescent protein) spectrum derived from ECFP-Ku70/Ku80. (**C**) Quantification of relative FRET efficiency after the addition of the indicated oligonucleotide substrates (access from the biotinylated T was blocked by the addition of streptavidin). Error bars depict the standard error of the mean (SEM). (**D**) Top: Schematic representation of the fluorescently tagged Ku70 and Ku80 polypeptides. Bottom: Cartoon representation of the Ku heterodimer (1JEQ.pdb [5]) with core Ku70 domain in pink, SAP domain in red, core Ku80 domain in blue, and the location of the Ku80 C-terminal extension indicated with the cyan sphere. DNA would be bound in the central cavity, resulting in structural rearrangements of SAP and C-terminal domains [21]. Model created with PyMol (version 2.3.3, Schrödinger LLC, New York, NY, USA).

**Figure 2 ijms-21-06725-f002:**
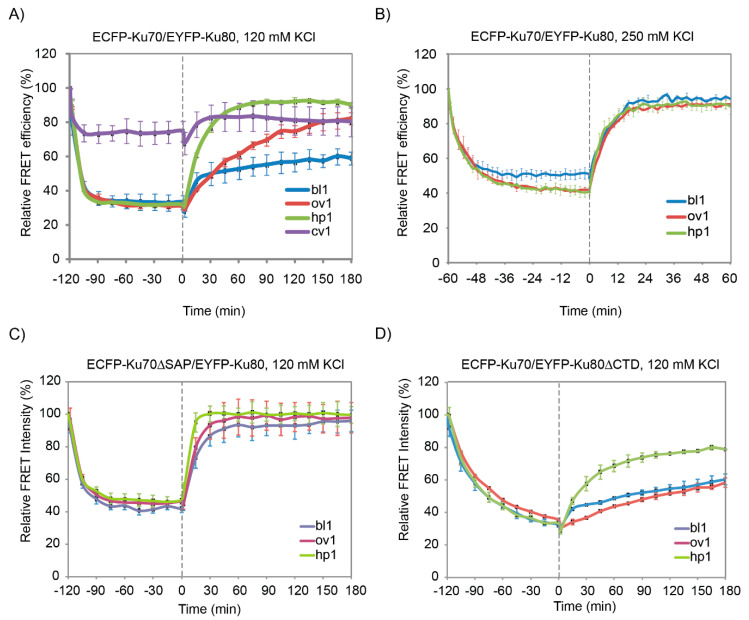
Recovery of the FRET signal after incubation of Ku-DNA complexes with untagged Ku70/80. (**A**) ECFP-Ku70/EYFP-Ku80 was first incubated with the indicated oligonucleotide substrates (between time points -120 and 0), after which a 3-fold excess of untagged Ku70/80 was added to the mixtures, and FRET efficiency was determined at the indicated time points. (**B**) The same analysis as in A, but with 250 mM KCl instead of 120 mM KCl (also note the different time scale). Error bars depict the SEM. (**C**) ECFP-Ku70ΔSAP/EYFP-Ku80 or (**D**) ECFP-Ku70/EYFP-Ku80ΔC was incubated with the indicated DNA substrates and challenged with excess untagged Ku70/Ku80 protein at time point 0.

**Figure 3 ijms-21-06725-f003:**
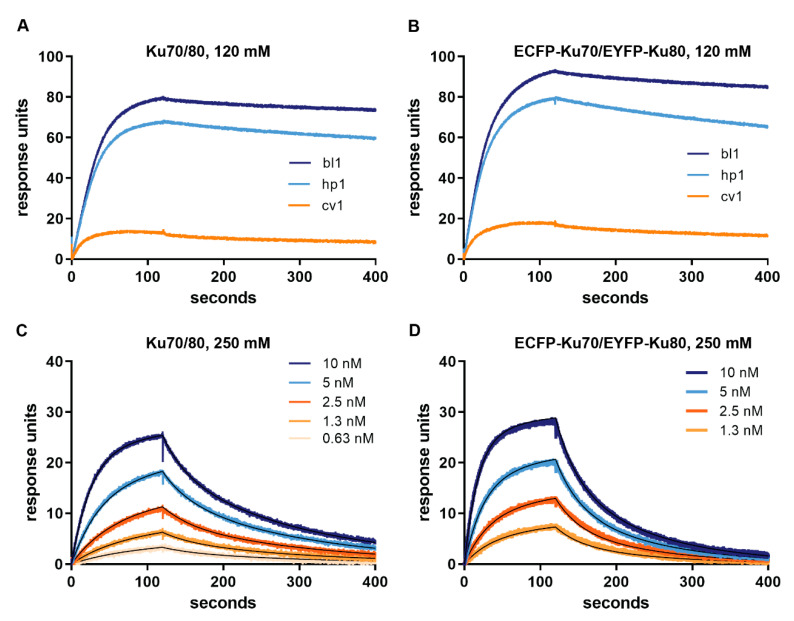
SPR (surface plasmon resonance) analysis of untagged Ku and ECFP-Ku70/EYFP-Ku80 binding to DNA. (**A**) Sensorgram of 10 nM Ku70/80 binding to blunt (bl1), hairpin (hp1), and clover-leaf (cv1) DNA at 120 mM NaCl. (**B**) Sensorgram of 10 nM ECFP-Ku70/EYFP-Ku80 binding to bl1, hp1, and cv1 DNA at 120 mM NaCl. (**C**) Sensorgram of Ku70/Ku80 (0.625–10 nM) binding to bl1 DNA at 250 mM NaCl. (**D**) Sensorgram of ECFP-Ku70/EYFP-Ku80 (1.25–10 nM) binding to bl1 at 250 mM NaCl. The fit of the model is indicated with thin black lines.

**Figure 4 ijms-21-06725-f004:**
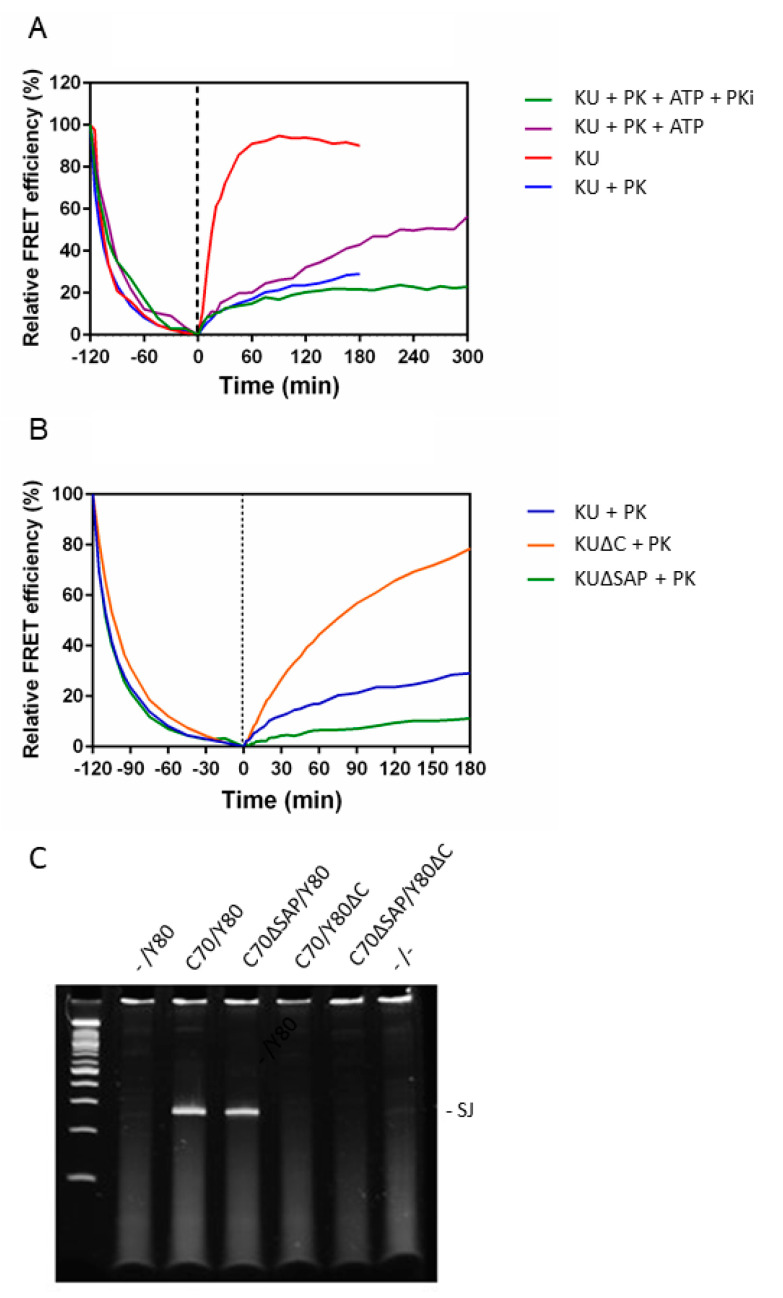
DNA-PK_CS_ (DNA-dependent protein kinase) stabilized Ku binding to DNA ends. (**A**) ECFP-Ku70/EYFP-Ku80 was incubated with DNA substrate bl1 in the presence or absence of DNA-PK_CS_ (PK) with or without ATP and the DNA-PK inhibitor Wortmannin (PKi), as indicated. (**B**) ECFP-Ku70/EYFP-Ku80 (KU), ECFP-Ku70ΔSAP/EYFP-Ku80 (KUΔSAP), or ECFP-Ku70/EYFP-Ku80ΔC (KUΔC) were incubated with bl1 and DNA-PKcs. In (**A**,**B**), the salt concentration was 250 mM KCl; the relative FRET efficiency was measured, and the excess untagged Ku70/Ku80 protein was added at time point 0. (**C**) V(D)J recombination activity in Ku70^−/−^/Ku80^−/−^ mouse embryonic fibroblasts complemented with ECFP-Ku70 (C70), EYFP-Ku80 (Y80), or mutants, as indicated. Signal joint (SJ) formation was tested by PCR amplification over the newly formed SJ.

**Table 1 ijms-21-06725-t001:** Binding of Ku70/80, ECFP-Ku70/EYFP-Ku80, and labeled Ku variants—Ku80ΔC, Ku70ΔSAP—to different DNA substrates (blunt bl1, overhang ov1, and hairpin hp1; see Figure 1A) at 250 mM NaCl quantified using surface plasmon resonance. Kinetic constants were derived by fitting a model to the data that accounts for binding of Ku to the free DNA end and a small fraction interacting with the chip matrix. *k*_a =_ association rate constant for binding of Ku to DNA; *k*_d_ = dissociation rate constant for release of Ku from DNA; *k*_f_ = forward rate constant for interaction with chip matrix; *k*_r_ = reverse rate constant for interaction with chip matrix; K_D_ = affinity constant.

Protein	Substrate	*k* _a_	*k* _d_	*k* _f_	*k* _r_	*K* _D_
		×10^6^ M^−1^ s^−1^	×10^-2^ s^−1^	×10^−3^ s^−1^	×10^−3^ s^−1^	nM
Ku70/Ku80	bl1	3.75	2.48	7.48	8.58	3.52
	ov1	3.44	2.48	4.59	7.50	4.47
	hp1	5.74	17.6	1.75	3.72	18.8
ECFP-Ku70/EYFP-Ku80	bl1	3.71	2.66	4.49	9.55	4.87
	ov1	3.77	3.43	3.42	9.01	6.60
	hp1	21.5	86.1	1.69	10.5	32.4
ECFP-Ku70/EYFP-Ku80ΔC	bl1	3.22	4.42	9.10	16.4	9.27
ECFP-Ku70ΔSAP/EYFP-Ku80	bl1	1.54	93.2	94.6	21.5	93.9

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
