# Peer review of "C-Terminal Extensions of Ku70 and Ku80 Differentially Influence DNA End Binding Properties"

_ijms, 2020, doi:10.3390/ijms21186725_

Round 1
Reviewer 1 Report
Review :
C-terminal extensions of Ku70 and Ku80 differentially influence DNA end binding properties, Inagawa T et.al.
The authors showed that (1) relative binding efficiencies of KU proteins to various DNA substrates using FRET analyses, (2) measured stabilities of KU binding to the substrates in the presence and absence of DNA-PKcs by FRET. The kinetic analyses of KU proteins with DNA-PKcs by FRET are new, however, the results shown here do not go much beyond what have been shown by the structural studies (below) and biochemical studies. Further experiments on different substrates and/or more mechanistic studies are suggested. Important text (for Fig 2) is missing. Presentation of data (i.e. Fig. 4) should be revised.
- Line 104- 105: it appears that the sentence is not completed, or somehow lost. Entirely missing explanation for Fig. 2.
- Why were mouse embryonic fibroblasts used for human KU complementation experiments? A human cell system with KU knockout or KU knockdown is more ideal.
- Do the KU70-SAP and KU80dC mutant proteins complement cNHEJ?
- A figure to show domains of KU proteins will be helpful to readers.
- Fig 4: Should be plotted with the wild-type protein data.
- Structural studies suggest the importance of the KU domain for interacting with DNA and DNA-PKcs (Yin X et al., Cell Res, 2017, Sharif H et al., PNAS, 2017, Wu Q et al., 2017 Prog Biophys Mol Biol, 2019). The authors do not cite or discuss the papers.
- Most of the results have been predicted by the structural papers mentioned above in (4), except that the authors observed KU80 interaction does not require the C-terminal domain, but has reduced affinity, and efficient interaction of KU to the hairpin DNA.
- Sup Fi. 2 lacks error bars.
- Efficient binding of KU to the hairpin structure is interesting although with lower affinity. It has been shown that KU binds to hairpin RNA (Dalby et al., RNA, 2013, Anisenko et al., Biochimie 2017). Do the KU complex and KU-DNA-PKcs have as strong affinity to RNA substrates? The involvement of RNA in DSB repair have been shown recently, and it is interesting to see the differences of KU affinities to DNA and RNA substrates with and without DNA-PKcs.
Minor comments:
- The fonts used in the text are not consistent.
Author Response
General response: All reviewers comment on a missing section of text and a missing figure legend 2. We think this may be an unfortunate event that occurred during editorial processing of our manuscript, as the two sections are actually present in the version we submitted, and are also present in the edited version that we received back from the journal, but in a different font. We also note that because of this, the line numbers do not correspond between our edited version and in the versions of the reviewers.
Reviewer 1
Line 104- 105: it appears that the sentence is not completed, or somehow lost. Entirely missing explanation for Fig. 2.
We apologize for the missing sections and think this may have been lost from our submitted version during the editorial process (see above). The revised manuscript is complete.
Why were mouse embryonic fibroblasts used for human KU complementation experiments? A human cell system with KU knockout or KU knockdown is more ideal.
This is a technical issue. Human Ku knock out cells are not viable, while mouse cells are. We considered knock down less desirable, as residual activity of wild type Ku protein could obscure effects of mutations. Furthermore, full complementation between human and murine Ku proteins and the constituting subunits has been shown in several publications over the past 20 years, so the simplest technical solution is murine cells with the human proteins expressed.
Do the KU70-SAP and KU80dC mutant proteins complement cNHEJ?
The most direct way to show complementation of cNHEJ appears to be the V(D)J recombination assay, as this is fully dependent on this repair pathway. Other assays show at least some backup of alternative end-joining pathways (e.g. ionizing radiation sensitivity or transfection of linear DNA into cells).
A figure to show domains of KU proteins will be helpful to readers.
We added a schematic of Ku (Figure 1D), showing the locations of the Ku70 and Ku80 core domains and C-terminal extensions.
Fig 4: Should be plotted with the wild-type protein data.
We thank the reviewer for this suggestion. We now merged figure 2 and 4 to clearly show wild type and mutant Ku proteins in the same figure (new figure 2 with panels C and D).
Structural studies suggest the importance of the KU domain for interacting with DNA and DNA-PKcs (Yin X et al., Cell Res, 2017, Sharif H et al., PNAS, 2017, Wu Q et al., 2017 Prog Biophys Mol Biol, 2019). The authors do not cite or discuss the papers.
References have been added.
Most of the results have been predicted by the structural papers mentioned above in (4), except that the authors observed KU80 interaction does not require the C-terminal domain, but has reduced affinity, and efficient interaction of KU to the hairpin DNA.
As mentioned above, these references have been added.
Sup Fi. 2 lacks error bars.
These experiments were carried out as single titration experiments to get a first indication whether physiological conditions were appropriate for the FRET assays. For this purpose single measurements were sufficient. In subsequent experiments everything was measured in triplicate.
Efficient binding of KU to the hairpin structure is interesting although with lower affinity. It has been shown that KU binds to hairpin RNA (Dalby et al., RNA, 2013, Anisenko et al., Biochimie 2017). Do the KU complex and KU-DNA-PKcs have as strong affinity to RNA substrates? The involvement of RNA in DSB repair have been shown recently, and it is interesting to see the differences of KU affinities to DNA and RNA substrates with and without DNA-PKcs.
We did not go into RNA binding, because we do not know how FRET efficiency would be influenced by RNA binding. It is an interesting question, but beyond the scope of this manuscript.
Minor comments:
The fonts used in the text are not consistent.
We think this occurred during editorial processing of our manuscript, and we assume that it will be corrected in the final version of the paper.

Reviewer 2 Report
The manuscript reports the dvelopment and application of a FRET based assay to study the kinetic and thermodynamic of the binding of Ku70/80 to different DNA oligomers. This physical process is indeed a first crucial step in double strand break repair and in particular in NHEJ pathway. Furthermore, the effect on the binding of the presence of DNA kinase and comparison with surface plasmon resonance results are provided.
The work is clearly sounding and interesting offering clear evidences on the DNA associations kinetics, and on the reversibility of the complex between DNA and the repair protein. It also highlights the complementarity and utility of FRET assays. The manuscript is globally well written and easy to follow. As such I recommend publication of the present contribution once the authors will have addressed some minor issues. Further review is not necessary.
-To facilitate the reading I would include a scheme in the Introduction reporting the different steps of the NHEJ pathways and specifically the role of Ku70/80 and DNA kinases that are further discussed.
-Figure 2 is not discussed in the text, and indeed section 2.3 seams incomplete.
-There is a repeated paragraph at the end pf page 7.
Author Response
Inagawa et al
General response: All reviewers comment on a missing section of text and a missing figure legend 2. We think this may be an unfortunate event that occurred during editorial processing of our manuscript, as the two sections are actually present in the version we submitted, and are also present in the edited version that we received back from the journal, but in a different font. We also note that because of this, the line numbers do not correspond between our edited version and in the versions of the reviewers.
Reviewer 2
The manuscript reports the dvelopment and application of a FRET based assay to study the kinetic and thermodynamic of the binding of Ku70/80 to different DNA oligomers. This physical process is indeed a first crucial step in double strand break repair and in particular in NHEJ pathway. Furthermore, the effect on the binding of the presence of DNA kinase and comparison with surface plasmon resonance results are provided.
The work is clearly sounding and interesting offering clear evidences on the DNA associations kinetics, and on the reversibility of the complex between DNA and the repair protein. It also highlights the complementarity and utility of FRET assays. The manuscript is globally well written and easy to follow. As such I recommend publication of the present contribution once the authors will have addressed some minor issues. Further review is not necessary.
-To facilitate the reading I would include a scheme in the Introduction reporting the different steps of the NHEJ pathways and specifically the role of Ku70/80 and DNA kinases that are further discussed.
As many good reviews have been written on this subject, we do not think it would add much here, as we only studied the first step in the reaction: binding to DNA ends.
-Figure 2 is not discussed in the text, and indeed section 2.3 seams incomplete.
We apologize for the missing sections and think this may have been lost from our submitted version during the editorial process. The revised manuscript is complete.
Figure 2A is discussed in section 2.3.
-There is a repeated paragraph at the end pf page 7.
Has been deleted now.
Reviewer 3 Report
The authors present a description of their study aimed at the analysis of Ku70/80 heterodimer interactions with various DNA substrates using FRET assay as well as Surface Plasmon Resonance. It is the FRET assay application that is the main point of this work, and for this the authors prepared Ku70 and Ku80 tagged with fluorescent proteins and showed that ECFP-Ku70/EYFP-Ku80 heterodimer is functional in DNA and DNA-PKCS binding. There are some interesting findings, for example, the authors showed that Ku70 SAP domain does not influence the kinetics of Ku dissociation from the whole complex DNA-PK-DNA complex. However, there are a lot of points in the article that need to be corrected or more thoroughly explained. In addition, the article is written carelessly with some text repetitions, and in section 2.3 there is a lack of the text.
Other points:
Introduction - lines 61-62: the phrase “We found that the Ku70 and Ku80 C-terminal domains have functions in DNA end binding and DNA-PKCS interaction, respectively” is not informative because it has been found some time ago, see review of Fell & Schild-Poulter in Mutation Research (2015), for example.
Line 86: “A mixture of ECFP-Ku70/Ku80 and Ku70/EYFP-Ku80 did not give a FRET signal” – it is strange that there is no exchange between subunits of these two complexes.
Lines 92-101: the description of DNA substrates is unclear. The bl1 and ov1 structures are called hairpins in the nucleic acid nomenclature, while the hp1 is called a dumbbell. What is a linker in hp1 and hp2?
Line 104: it is impossible to understand this section because of the lack of the text.
Line 130: what is dissociation rate constants for bl1?
Line 133: how were dissociation rate constants determined in the FRET experiments?
Figure 3. What are E and F?
Line 159: The contribution of the Ku70 SAP domain to its DNA binding properties has been shown in Anisenko et al, Biochimie 2017, its contribution to DNA binding properties of the Ku heterodimer was shown at least in Wang et al, J. Biol. Chem. (1998), and Rivera-Calzada et al, EMBO Rep. 8 (2007). These works should be cited.
Table 1 and lines 163-166: abbreviations Ku80ΔC tagged, Ku70ΔSAP tagged and ΔSAP variant are inappropriate, since they do not allow to understand what is discussed, a monomer or dimer
Lines 190-196: The text is unclear and should be verified and re-written.
Lines 206-209 repeat lines 202-205.
Lines 263-264: DNA-PK complexes with the Ku80 C-terminal deletion mutant are less stable than wild type complexes, but more stable than Ku alone. Does it mean that the DNA-PK complex containing Ku80ΔC is more stable than the Ku heterodimer without DNA and DNA-PKcs? Moreover, this type of statement requires precise quantitative data that are not available in the article.
Author Response
Inagawa et al
General response: All reviewers comment on a missing section of text and a missing figure legend 2. We think this may be an unfortunate event that occurred during editorial processing of our manuscript, as the two sections are actually present in the version we submitted, and are also present in the edited version that we received back from the journal, but in a different font. We also note that because of this, the line numbers do not correspond between our edited version and in the versions of the reviewers.
Reviewer 3
The authors present a description of their study aimed at the analysis of Ku70/80 heterodimer interactions with various DNA substrates using FRET assay as well as Surface Plasmon Resonance. It is the FRET assay application that is the main point of this work, and for this the authors prepared Ku70 and Ku80 tagged with fluorescent proteins and showed that ECFP-Ku70/EYFP-Ku80 heterodimer is functional in DNA and DNA-PKCS binding. There are some interesting findings, for example, the authors showed that Ku70 SAP domain does not influence the kinetics of Ku dissociation from the whole complex DNA-PK-DNA complex. However, there are a lot of points in the article that need to be corrected or more thoroughly explained. In addition, the article is written carelessly with some text repetitions, and in section 2.3 there is a lack of the text.
We apologize for the missing sections and think this may have been lost from our submitted version during the editorial process. The revised manuscript is complete.
Other points:
Introduction - lines 61-62: the phrase “We found that the Ku70 and Ku80 C-terminal domains have functions in DNA end binding and DNA-PKCS interaction, respectively” is not informative because it has been found some time ago, see review of Fell & Schild-Poulter in Mutation Research (2015), for example.
We don’t want to claim that we found this function. Therefore, we changed this sentence to: ‘We further defined the function of the Ku70 and Ku80 C-terminal domains in DNA end binding and DNA-PKCS interaction.’
Line 86: “A mixture of ECFP-Ku70/Ku80 and Ku70/EYFP-Ku80 did not give a FRET signal” – it is strange that there is no exchange between subunits of these two complexes.
The Ku70/80 heterodimer is so stable that exchange does not occur within the time frame of the experiment to a measurable extent. This is not unexpected, as both subunits have a very large and intertwined interface (see also the new figure 1D).
Lines 92-101: the description of DNA substrates is unclear. The bl1 and ov1 structures are called hairpins in the nucleic acid nomenclature, while the hp1 is called a dumbbell. What is a linker in hp1 and hp2?
We described the substrates according to the configuration of their binding end. It is correct, that technically speaking, bl1 and ov1 are hairpins, but we termed them blunt and overhang, because of the free end that is blunt or with a 3 base overhang. Similarly, the dumbbell substrate was termed hairpin, because the binding end forms a hairpin, in which the top strand is covalently coupled to the bottom strand. The end with the biotinylated T is blocked by streptavidin binding, so it is not really relevant whether this is covalently coupled between top and bottom strand.
We are not completely sure what the reviewer means by ‘the linker’. We assume that it is the linking half circle between top and bottom strand; this is just a 3’-OH of the top strand to the 5’-phosphate of the bottom strand.
Line 104: it is impossible to understand this section because of the lack of the text.
We apologize for the missing sections and think this may have been lost from our submitted version during the editorial process. The revised manuscript is complete.
Line 130: what is dissociation rate constants for bl1?
The value of the dissociation rate constant is stated in the same sentence (which is line 139-140 in our version) The meaning of the constant is the rate constant with which the protein dissociates from the DNA according to a single exponential decay function. We have added this to the Materials and Methods section.
Line 133: how were dissociation rate constants determined in the FRET experiments?
We applied a simple exponential as the model, this has now been described in the methods section.
Figure 3. What are E and F?
Unfortunately we submitted an old version of Figure 3. We apologize sincerely and have now included the correct version, which does not contain panels E and F because they had been included already in supplemental figure S3.
Line 159: The contribution of the Ku70 SAP domain to its DNA binding properties has been shown in Anisenko et al, Biochimie 2017, its contribution to DNA binding properties of the Ku heterodimer was shown at least in Wang et al, J. Biol. Chem. (1998), and Rivera-Calzada et al, EMBO Rep. 8 (2007). These works should be cited.
Papers have been cited.
Table 1 and lines 163-166: abbreviations Ku80ΔC tagged, Ku70ΔSAP tagged and ΔSAP variant are inappropriate, since they do not allow to understand what is discussed, a monomer or dimer
They have been renamed to the nomenclature used in the rest of the paper.
Lines 190-196: The text is unclear and should be verified and re-written.
I am not sure which lines the reviewer refers to. Line numbers have shifted in the editorial version we received. Maybe the reviewer can clarify what is unclear to him/her?
Lines 206-209 repeat lines 202-205.
Has been corrected.
Lines 263-264: DNA-PK complexes with the Ku80 C-terminal deletion mutant are less stable than wild type complexes, but more stable than Ku alone. Does it mean that the DNA-PK complex containing Ku80ΔC is more stable than the Ku heterodimer without DNA and DNA-PKcs?
No, we mean that it is more stable than wild type KU on DNA without DNA-PKcs. To make this clear we have rephrased the sentence to:
DNA-PK complexes on blunt DNA ends with the Ku80 C-terminal deletion mutant are less stable than wild type complexes, but more stable than Ku without DNA-PK.
Moreover, this type of statement requires precise quantitative data that are not available in the article.
The statement follows from the data in figure 5A and B (figures 4A and B in the revised version), for which we have not quantified the rates. We think the observed differences in panels A and B are so clear that we can make this statement without quantitation.
Reviewer 4 Report
Manuscript ID: ijms-879334
Type of manuscript: Article
Inagawa et al., “C-terminal extensions of Ku70 and Ku80 differentially influence DNA end binding properties”.
This work present a new approach based on FRET, to study protein/protein interaction with different DNA.
1. Manuscript preparation problems
Police heterogeneity lines 132 and 133, lines 166 and 167, line 198 to 209 are not with the same police as the rest of the document.
Missing /duplicated sentences.
-In the result section, 2.3 There is only two sentences and the last sentence (line 104) is incomplete. Therefore, the results description of this section is completely missing.
-Repeated sentences in the discussion part lines 202 to 204 are repeated at lanes 206 to 208…
Figures
-The figure 3 caption is incomplete. We have no information about the E and F graphics. These are not even cited in the results section.
-The figure 5 caption is non homogenous. DNA-PK or PK are used alternatively one the same figure.
-Poor quality of the figure 1. The red sign is not clear at all.
-The figure 5 graphic. For some experiments the dissociation time is 180 min and for other is 300 minutes. Why ? It would be better to homogenize that.
-Figure 4: The description of the different nucleic acids used in the experiment has to be done for the figure 4B too.
Supplemental figure 1B: it s very difficult to analyse it since there is no information about all the wells content. What was loaded on the 4th well? If it is 75ng of KU 70/80 proteins, why there is no more free DNA? In EMSA experiments, DNA must be in large excess to ensure the presence of free DNA in all the experiments.
References:
-References are not always the good ones. (ref 13 and 16) when a work is cited, the princeps article describing the work has to be cited and not a review.
2. Core problems:
Results interpretations:
Double-tagged Ku heterodimer
The results obtained with the WT double-tagged Ku heterodimer are presented on the figure 2A. The results obtained for the double-tagged mutKu70/wtKu80 or the double-tagged wtKu70/mutKu80 heterodimers are presented on figure 4 A and 4B respectively. In these experiments after the association of the different Double-tagged Ku heterodimer, the dissociation has been followed by WT unlabeled Ku heterodimer addition. So the observed differences give information about the impact of the mutations on the stability of the Double-tagged Ku heterodimers.
After putting your results obtained with the different Ku proteins on the same graph, with one graph per substrate, i have some questions.
-Double-tagged mutKu70/wtKu80 heterodimer: there is a difference for all the DNA substrate tested and you conclude that the SAP domain is essential for stable binding to DNA and i agree with that.
-Double-tagged wtKu70/mutKu80 heterodimer: It’s clear that for bl1 substrate, there is no difference between Wt or mutated Ku80. But there is a big one with the Ov1 substrate and you conclude in lines 170 and 171 that “the dissociation is not affected by this mutation” (the Ku80 C-ter deletion). I dont’t agree with your interpretations. There are some interesting differences between the DNA substrates. Why this aspect is not investigated?
In the conclusion part, you conclude that the Ku80 Cter domain stabilizes the DNA-PK holocomplexe. This conclusion is valid only for the bl1 substrate (witch gives no difference between the double Wt heterodimer and the mut80 one). The conclusion should be modified according to the results by signaling the DNA substrate effect.
FRET results comparison to the SPR data:
In the supplemental data figure 3B you present the association/dissociation of the wt double-tagged Ku70/Ku80 heterodimer. In E and F you present the double-tagged mutKu70/wtKu80 heterodimer and the double-tagged Ku70/mutKu80 heterodimer respectively.
Comparison between B and E, (5 fold more protein and two fold less response unit) indicates that the SAP domain of Ku70 is indeed implicated in the bl1 binding. This result conforms with your FRET based results.
Comparison between B and F, (2,5 fold more protein and approximatively the same response unit) indicates a role of the Cter domain of Ku80 in Bl1 binding. This point is not congruent with your FRET based conclusions. This point should be addressed and discussed.
Author Response
Inagawa et al
General response: All reviewers comment on a missing section of text and a missing figure legend 2. We think this may be an unfortunate event that occurred during editorial processing of our manuscript, as the two sections are actually present in the version we submitted, and are also present in the edited version that we received back from the journal, but in a different font. We also note that because of this, the line numbers do not correspond between our edited version and in the versions of the reviewers.
Reviewer 4
This work present a new approach based on FRET, to study protein/protein interaction with different DNA.
- Manuscript preparation problems
Police heterogeneity lines 132 and 133, lines 166 and 167, line 198 to 209 are not with the same police as the rest of the document.
Font heterogeneity was probably introduced during editorial processing and we assume this will be corrected in the final version; this is beyond control of the authors.
Missing /duplicated sentences.
-In the result section, 2.3 There is only two sentences and the last sentence (line 104) is incomplete. Therefore, the results description of this section is completely missing.
We apologize for the missing sections and think this may have been lost from our submitted version during the editorial process. The revised manuscript is complete.
-Repeated sentences in the discussion part lines 202 to 204 are repeated at lanes 206 to 208…
Duplicated sentences have been deleted.
Figures
-The figure 3 caption is incomplete. We have no information about the E and F graphics. These are not even cited in the results section.
Unfortunately we submitted an old version of Figure 3. We apologize sincerely and have now included the correct version, which does not contain panels E and F because they had been included already in supplemental figure S3.
-The figure 5 caption is non homogenous. DNA-PK or PK are used alternatively one the same figure.
We would like to thank the reviewer for bringing this to our attention: the figure has been adapted.
-Poor quality of the figure 1. The red sign is not clear at all.
All figures have now been uploaded in high resolution.
-The figure 5 graphic. For some experiments the dissociation time is 180 min and for other is 300 minutes. Why ? It would be better to homogenize that.
Standard incubation time was 180 minutes. In order to make the difference between with and without DNA-PK inhibitor more clear, we extended this to 300 minutes for these two cases. This extension is only to make clear that there is a real difference between both situations.
-Figure 4: The description of the different nucleic acids used in the experiment has to be done for the figure 4B too.
We thank the reviewer for the suggestion. This figure has now been merged with figure 2 and description of substrates is in each panel.
Supplemental figure 1B: it s very difficult to analyse it since there is no information about all the wells content. What was loaded on the 4th well? If it is 75ng of KU 70/80 proteins, why there is no more free DNA? In EMSA experiments, DNA must be in large excess to ensure the presence of free DNA in all the experiments.
This figure is merely to show that both protein preparations form complexes of one and two Ku heterodimers onto the double stranded DNA. We don’t want to analyze binding affinity here, so we did not go into any detail in the analysis of the figure. We added a few words to the legend to clarify the content of the lanes. More precise measurements were done by SPR and FRET analysis.
References:
-References are not always the good ones. (ref 13 and 16) when a work is cited, the princeps article describing the work has to be cited and not a review.
Although we agree that one should preferentially refer to the original literature. However, in some cases it is less useful to put in many references when a certain aspect is discussed well in a review. We refer to the primary literature when we discuss how our results connect to the literature. For general description of the background, reviews are also used in some cases.
- Core problems:
Results interpretations:
Double-tagged Ku heterodimer
The results obtained with the WT double-tagged Ku heterodimer are presented on the figure 2A. The results obtained for the double-tagged mutKu70/wtKu80 or the double-tagged wtKu70/mutKu80 heterodimers are presented on figure 4 A and 4B respectively. In these experiments after the association of the different Double-tagged Ku heterodimer, the dissociation has been followed by WT unlabeled Ku heterodimer addition. So the observed differences give information about the impact of the mutations on the stability of the Double-tagged Ku heterodimers.
After putting your results obtained with the different Ku proteins on the same graph, with one graph per substrate, I have some questions.
-Double-tagged mutKu70/wtKu80 heterodimer: there is a difference for all the DNA substrate tested and you conclude that the SAP domain is essential for stable binding to DNA and i agree with that.
-Double-tagged wtKu70/mutKu80 heterodimer: It’s clear that for bl1 substrate, there is no difference between Wt or mutated Ku80. But there is a big one with the Ov1 substrate and you conclude in lines 170 and 171 that “the dissociation is not affected by this mutation” (the Ku80 C-ter deletion). I dont’t agree with your interpretations. There are some interesting differences between the DNA substrates. Why this aspect is not investigated?
We agree with the reviewer that our statements were a bit too strong here. We changed the sentence to: Dissociation was not severely affected by this mutation, although some differences were apparent for the ov1 substrate.
In the conclusion part, you conclude that the Ku80 Cter domain stabilizes the DNA-PK holocomplexe. This conclusion is valid only for the bl1 substrate (witch gives no difference between the double Wt heterodimer and the mut80 one). The conclusion should be modified according to the results by signaling the DNA substrate effect.
We added the words ‘blunt’ and ‘on blunt DNA ends’ to specify our conclusion.
FRET results comparison to the SPR data:
In the supplemental data figure 3B you present the association/dissociation of the wt double-tagged Ku70/Ku80 heterodimer. In E and F you present the double-tagged mutKu70/wtKu80 heterodimer and the double-tagged Ku70/mutKu80 heterodimer respectively.
Comparison between B and E, (5 fold more protein and two fold less response unit) indicates that the SAP domain of Ku70 is indeed implicated in the bl1 binding. This result conforms with your FRET based results.
Yes indeed, and very clearly also based on the affinity constants as listed in table 1 (4.87 nM for wild type, 93.9 nM for SAP mutant. This is an almost 20-fold difference, big enough to appreciate from sensorgrams with different protein concentrations.
Comparison between B and F, (2,5 fold more protein and approximatively the same response unit) indicates a role of the Cter domain of Ku80 in Bl1 binding. This point is not congruent with your FRET based conclusions. This point should be addressed and discussed.
This difference is much smaller and difficult to fully appreciate from the sensorgrams. From the quantified affinities in table 1 it can be appreciated that the difference in affinity is actually not big, less than 2-fold (as opposed to almost 20-fold for the SAP mutant) and thus similar to the FRET experiment.
Round 2
Reviewer 3 Report
The article may be accepted for publication
Reviewer 4 Report
The manuscript has been improved. All the repeated sentences, missed parts corrected. Analysis of the results is more clearly presented.